# Improving the Head Pose Variation Problem in Face Recognition for Mobile Robots

**DOI:** 10.3390/s21020659

**Published:** 2021-01-19

**Authors:** Samuel-Felipe Baltanas, Jose-Raul Ruiz-Sarmiento, Javier Gonzalez-Jimenez

**Affiliations:** Machine Perception and Intelligent Robotics Group (MAPIR), Department of System Engineering and Automation, Biomedical Research Institute of Malaga (IBIMA), University of Malaga, 29071 Málaga, Spain; jotaraul@uma.es (J.-R.R.-S.); javiergonzalez@uma.es (J.G.-J.)

**Keywords:** face recognition, assistant mobile robots, cross-pose face recognition, MAPIR Faces, human-robot interaction

## Abstract

Face recognition is a technology with great potential in the field of robotics, due to its prominent role in human-robot interaction (HRI). This interaction is a keystone for the successful deployment of robots in areas requiring a customized assistance like education and healthcare, or assisting humans in everyday tasks. These unconstrained environments present additional difficulties for face recognition, extreme head pose variability being one of the most challenging. In this paper, we address this issue and make a fourfold contribution. First, it has been designed a tool for gathering an uniform distribution of head pose images from a person, which has been used to collect a new dataset of faces, both presented in this work. Then, the dataset has served as a testbed for analyzing the detrimental effects this problem has on a number of state-of-the-art methods, showing their decreased effectiveness outside a limited range of poses. Finally, we propose an optimization method to mitigate said negative effects by considering key pose samples in the recognition system’s set of known faces. The conducted experiments demonstrate that this optimized set of poses significantly improves the performance of a state-of-the-art, cutting-edge system based on Multitask Cascaded Convolutional Neural Networks (MTCNNs) and ArcFace.

## 1. Introduction

The application of robotics in human-populated environments in most cases requires the interaction between these robotic agents and the individuals present. The complexities of these interactions vary from one application to another; for example, a Roomba cleaning robot only needs to avoid individuals, whereas an assistant robot needs to give and receive feedback from the users [1]. As robots are demanded to accomplish more complex interactions, they need and increasing amount of additional information about the context in which they reside. The identity of each human in the environment is one of such pieces of information that unlock different options for human-robot interaction (HRI). Not only is it crucial in some settings, such as reminding an elderly user to take its medicine at a certain time, but also it can enrich everyday interactions by providing a customized service [2].

Human identification has been accomplished using various sources of biometric information; such as fingerprints, irises, voices, or faces [3]. Although all of these methods can be used to identify humans in a variety of settings, in the context of HRI, for example, the use of mobile robots in home environments, significantly favours the use of face recognition (FR) for user identification. In comparison to some of these methods, the standout features of FR are:It requires minimal interaction from the user as long as the robot has a sufficiently clear view of their face. This feature is imprescindible in HRI to achieve more natural interactions, not relying as much on human cooperation. In comparison, other methods are more intrusive on the user’s daily lives, for example, voice recognition always requires the user to speak, and fingerprint recognition requires the user to physically touch a sensor.The only peripheral needed for FR is a camera, which is a common component in most mobile robots.

FR performance has drastically improved in recent years, in response to advances in machine learning; namely the invention of convolutional neural networks (CNNs), and the widespread increase in the compute power [4,5]. These improvements in the state-of-the-art have been recorded according to various popular benchmark datasets like LFW [6], or IJB-A  [7]. The usual FR pipeline (Figure 1) is composed of a CNN for face detection, an optional face alignment step to mitigate pose variance, another CNN to extract deep feature embeddings from a face image, and a classification method using a comparison function between embeddings. Currently, some successful applications of FR are authentication in smartphones and identification in security cameras. Nevertheless, its use in uncontrolled environments presents detrimental conditions that can seriously hamper the performance of FR systems, such as: increased variability in the head pose, greater exposure to detrimental or changing illumination conditions, unpredictable user movements, and so forth [8,9,10].

Extreme pose variability is one of the hardest challenges in FR, as the appearance of the subject can vary drastically from two different perspectives. State-of-the-art algorithms such as FaceNet [11], ArcFace [12], and CosFace [13] still have difficulties to generalize from frontal faces to profile faces. Figure 2 depicts a real example in which ArcFace shows sub-human performance due to pose variations. In the context of HRI, these types of errors reduce considerably the usability of FR, since the system can only be trusted for quasi-frontal face images. In the case of a mobile robot equipped for HRI such as the Giraff robot [14], obtaining these types of faces is not impossible, but it requires additional effort:Since the robot is mobile, it can reposition itself to find the right perspective of the face. This is not always possible due to obstacles and requires the robot to estimate the pose of the head.Since the robot is equipped with speakers to interact with humans, it can talk with the user in an effort to get a frontal face image. This is more intrusive in the users’ daily lives, and if these occurrences are too common they might annoy the users.

The pose variability problem originates in the fact that training datasets are composed mostly by frontal images. Deep learning (DL) techniques require tremendous amounts of data—on the order of 1 million samples. This hunger of data greatly favours automated methods of image collection from Internet sources in favor of other, more manual, methods. Internet sources are not without bias, since most of them some common characteristics: depict famous individuals (introducing a source of possible age, race, and gender bias), show mostly frontal faces, are of professional quality (high resolution and good illumination conditions), and so forth. In comparison, obtaining samples with more varied head poses is considerably more expensive, since there are not many available sources.

In this paper, we further study the effects of adverse head poses to FR, one of the most detrimental conditions present in uncontrolled environments, such as identification for HRI. Then we propose a technique for mitigating the effect of this condition, making a fourfold contribution. First, we provide a pose-aware face collection tool to create datasets for face recognition (see Section 3). This face collection tool solves the issue of collecting face images with increased pose variance from a single individual. Since it only relies on a webcam and various state-of-the-art CNNs, it can be more widely distributed to facilitate the collection process. Second, we contribute of an evaluation dataset (this dataset is available at https://doi.org/10.5281/zenodo.4311764) for face recognition (Section 4). The dataset contains a uniform distribution of poses across pitch and yaw rotation angles. This feature makes it an ideal benchmark to analyze the effects of pose variance in FR algorithms. Then, we also contribute evaluations of multiple state-of-the-art neural networks for face recognition on said dataset. This evaluation puts particular emphasis on discerning the effects adverse head poses on the algorithms’ performances (Section 5). Finally, we study how face recognition metrics improve across the head pose space with the increase of head pose variance in the gallery set at the classification step (Section 6). This study assesses which poses in the gallery set are most relevant. The selection of these poses is automated using an optimization method, analyzing how the performance metrics improve as more images are included in the gallery set As an illustrative result on our novel MAPIR Faces dataset, ArcFace shows an accuracy improvement from 0.73 to 0.95 by increasing the gallery dataset from 1 to 5 distinct poses per individual. This distinct set of poses are chosen using the proposed optimization method to maximize the accuracy of the system.

## 2. Related Works

This section provides a brief overview of each of the topics covered in this work. Although FR is the principal problem addressed, the proposed method heavily relies on head pose estimation (HPE) techniques. Additionally, the development of the MAPIR Faces dataset is underpinned with a study of contemporaneous face datasets, particularly those that approach the pose variance problem—either by including more variety of poses or directly including pose information for each image. Therefore, this section covers the following topics: face recognition (Section 2.1), head pose estimation (Section 2.2), and face recognition datasets (Section 2.3).

### 2.1. Face Recognition

Face recognition (FR) has been one of the most active research topics in the field of computer vision [15,16] from as early as the 1990s. One of the first major historical approaches was that of Eigenfaces [17] which, similarly to other approaches in the 1990s [18], used holistic methods to extract low-dimensional representations from the whole image. These first approaches, although achieving some deal of success in controlled environments, were promptly faced with some great challenges. The pose-illumination-expression (PIE) problem in FR was known as early as 1964 [19], and yet it remains as one of the greater issues in the field.

Since then the field has evolved drastically, since the tremendous rise in compute power has enabled the development of new algorithms [5,11,12]. Specially, the advent of DL and CNNs have brought on human-like performance in ideal conditions [4,5]. The newest advances in the field have originated in the development and application of new CNN techniques. Some of these advancements have been: the aggregation of massive datasets for training CNNs [20,21,22,23], the development of new loss functions to improve feature extraction [11,12,13], or the use of novel neural network architectures [24,25,26]. The problem of head pose variance in FR systems has been addressed in other works: Huang et al. [27] applied generative adversarial networks (GANs) to synthesize frontal faces before feature extraction and Cao et al. [28] used estimated poses to approximate a deep embedding of the corresponding frontal face. Although these methods obtain increased resilience to pose variance, they do so by relying on additional and often more computationally expensive modules at inference time, such as GANs or secondary CNNs. Comparatively, few efforts have been carried out to understand the effects of the gallery dataset on the recognition process.

### 2.2. Head Pose Estimation

Head pose estimation (HPE) is a challenging regression task that requires inferring 3D information from 2D images. It has been a topic of interest in computer vision for a long time, since it can be used to detect the gaze and the attention of humans. Some approaches for HPE have been landmark-based [29,30,31], parametrized appearance models [32,33], and DL models [34,35,36]. Since the advent of CNNs, DL models have become one of the predominant avenues in HPE. HPE methods usually follow one of two approaches: they regress the pose directly from a face image, or they locate facial keypoints in the image—using those to infer the head’s pose. This latter approach, despite the existence of very accurate landmark detectors, presents some difficulties such as: an inability to infer poses if a high enough number of landmarks are missing or occluded, and their reliance on 3D head models. Although advancing the state-of-the-art in HPE is outside the confines of this work, HPE is a core component of the face collection application described in Section 3.2, since it will be used to select the images to be stored in the gallery set.

### 2.3. Related Datasets

This section reviews some of the recent face datasets, particularly those that contain pose-related information. The datasets considered include Biwi Kinect Head Pose Database [37], University of Cordoba Head database (UcoHead) [38], and University of Maryland Faces database (UMDFaces)  [39], among others:Biwi Kinect Head Pose Database [37]:
This repository has been a major source of inspiration for this paper. It contains a large amount of images per individual (∼500) in various poses, each annotated with their corresponding Euler angle. The major downside of this dataset is the fact that the poses available for each individual vary greatly from each other, lacking some of the more extreme poses for most individuals.
UcoHead dataset [38]:
Dataset that shares many commonalities with our desired dataset: it provides many images from a set of individuals and a uniform distribution of pitch and yaw poses. Despite all these similarities, some of its characteristics makes it unfit to analyze FR algorithms, namely: the low resolution of its images (40×40), and the fact that they are all grayscale.
UMDFaces dataset [39]:
One of the largest datasets available with fine-grained pose annotations. Nevertheless, they are gathered from public media (e.g., newspapers, internet, etc.), and therefore the poses available for each individual vary considerably from one another.


The remaining datasets analyzed in this work do not provide any pose-related properties relevant for this task. For example, IJB-A [7], Cross-Pose Labeled Faces in the Wild (CPLFW) [8] only provide extended head pose variability compared to previous datasets, while 300W-LP [40], Bosphorus [41] have a small set of poses annotated. Since these datasets lacked many of the properties required to evaluate the effects of head pose variance in FR algorithms, the MAPIR Faces dataset was developed to fill such a role. The details of this dataset are covered in Section 4.

## 3. Face Collection Tool

The lack of head pose variability in most face datasets is not easily solved by collecting more images from the Internet. Individual image collection, despite being more expensive, allows the creation of more varied datasets. These datasets are not only concerned with the amount of images they contain, but also with the representation of detrimental conditions such as pose, illumination or expression changes. The scope of this paper is limited to increase head pose variability in face datasets—one of the most common detrimental factors encountered by mobile robots in uncontrolled environments. In order to create a dataset containing a more diverse and uniformly distributed set of head poses, an application must meet two requirements: it must be able to infer the head pose of the user (see Section 3.1) and it must provide an interface to guide the movements of the user (Section 3.2).

### 3.1. Head Pose Estimation

Fast and reliable HPE is the central pillar of the face collection tool. The application must estimate the head pose of the user from a constant stream of intensity images at real-time speeds. These estimations are used to store poses of interest and give appropriate feedback to the user.

DL has become one of the predominant techniques for HPE in recent years, emulating the success of other computer vision problems such as object detection, or FR [4,5]. In this work, a number of open-source state-of-the-art algorithms for HPE have been tested on evaluation datasets and during our practical experiences developing the application:3D Dense Face Alignment (3DDFA) [34]
is a pose estimation method that attempts to fit a 3D morphable face model using a cascaded CNN and a set of face landmarks. A Pytorch implementation of the method described is provided as a GitHub repository (https://github.com/cleardusk/3DDFA). This implementation iterates over the original paper and provides various models pretrained on the 300W and 300W-LP datasets [34,40].
Hopenet [35]
is a CNN based HPE method which aims to compute the 3D pose components directly from an RGB image instead of estimating face landmarks as an intermediate step. Some interesting contributions in this paper are: use of multiple loss functions (one per angle component); and a study on the effects of low resolution images and the usage of landmarks in pose estimation. A public implementation is available at Github (https://github.com/natanielruiz/deep-head-pose), which contains some models pre-trained on 300W-LP.
FSA-Net [36]
is one of the most recent HPE publications available. The authors introduce some interesting ideas to the field of HPE, e.g., they borrow some ideas from age estimation systems, and they use fine-grained spacial structures. At the time of this work, they allegedly surpass most state-of-the-art methods in HPE benchmarks such as Biwi [37] and AFLW-2000 [34]. A public implementation is available in Github (https://github.com/shamangary/FSA-Net), which also contains some models pre-trained on 300W-LP and Biwi.


Although obtaining perfect head pose information is not the main objective of this application, the accuracy of the estimation method still has huge importance in the gathering process. Using inaccurate or unreliable estimation methods can severely hinder the process. In the case the estimator mistakenly outputs of the poses that maximizes the interest function of the system, it will not be able to correct this error without human intervention. During empirical testing of the application, it was found that FSA-Net, theoretically the most accurate of the methods reviewed, was not enough for our purposes. Estimations on the most extreme yaw poses (around ±55°) caused some significant issues which discouraged its use for the dataset creation process.

As 3 HPE methods already tested and ready to be integrated into the application, the elaboration of a stacking ensemble of these methods was considered the best possible solution to improve the performance. Each face image detected is fed to the three estimation methods in parallel as seen in Figure 3. Once the three pose estimations are computed, their mean is used as output of the system. Averaging the output of multiple estimators is a common tool to reduce variance for most DL methods [42]. This solution is by no means perfect, since it increases the resources needed to run the application and reduces the rate at which estimations are computed. However, the results from the stacking estimator—both in the aforementioned BIWI dataset and during practical testing—improved significantly, mitigating the undesired behaviors of the individual estimators to a great extent, e.g., unstable predictions, lack of repeatability for similar conditions, and a reduced effective range of poses.

Table 1 shows the evaluation results for each of the public implementations and the stacking estimator on the BIWI dataset. The base estimators all have considerable amounts of error on this dataset, enough to difficulty the data collection. Although 3DDFA performs better than others, still has considerable amount of error across all three angle components. The stacking classifier, which is composed of all three others, performs considerably better than each of the individual methods—specially in the pitch and yaw predictions. On the other hand, Table 2 shows the trade-off between the execution of all three methods and the stacking estimator. As stated before, the use of stacking estimators greatly increases the computational cost of the system. Nevertheless, it was deemed necessary for the correct functioning of the application, even at the cost of a slower collection process (∼4 Hz running on CPU and ∼25 Hz on GPU).

### 3.2. Interactive Application

A desktop application was considered the most appropriate approach to deploy this face collection tool. It contains a GUI to streamline the face collection process and increase the pose variance of said face samples. It accomplishes this task by continuously computing the head pose of the user in real-time. Then, the system selects which estimations are stored according to a predefined pose criteria. The GUI is updated at each step to give the user feedback about the current estimated pose and completion progress. This section describes the face collection process and showcases the GUI application.

Log-in.
As the user starts the application, they are prompted to follow a number of initial configuration steps: entering an identifier, selecting a storage location, and choosing a camera; as seen in Figure 4. The application has been developed with the usage of an RGB-D camera in mind. At the time of this writing, the application supports the usage of the Orbecc Astra RGB-D camera through the Openni library (https://github.com/severin-lemaignan/openni-python). Additionally, RGB camera support is provided via the OpenCV library (https://opencv.org). This enables a greater number of devices to use the application, since RGB-D cameras have yet to receive a widespread adoption.
Image collection.
The user is presented with the main view of the application (Figure 5). This view contains three main components for the user to control and receive feedback about the collection process—the collection state (Figure 5①), the camera feed (Figure 5②) and the control buttons (Figure 5③). The camera feed and the control buttons are very straightforward: the former shows the video feed provided by the camera, while the latter is used to pause or finish the collection process. The collection state contains a 2D grid (Figure 5①) which divides evenly a pitch-yaw space—the yaw ranges between (−65.0,65.0) and the pitch ranges between (−35.0,35.0). The limits of this 2D space have been defined in consideration towards preserving the accuracy of the methods selected in Section 3.1, as both their accuracy and stability decay significantly outside these bounds. A black pointer (Figure 5④) shows the current estimation provided by the system in real-time. The images stored in the dataset are chosen in consideration of their estimated head pose. Currently, the application stores a single image for each cell. The distance from the current yaw-pitch pose to the center of the cell is used as a scoring function. The nearer a pose is to the center, the more useful it is considered to the system. In this way, the color of each cell represents the value of this score at the current moment. The colorbar on the right (Figure 5⑤) shows the user the range of scores, where the blue colored cells represent that a pose close to the center of each cell is already stored within the system.


After the users have traversed the whole space, they can finish the collection process manually to store all the images and auxiliary data to the disk. The application does not send any data automatically. In the context of this work, the users sent said data to the authors in order to incorporate them to the repository described next.

## 4. MAPIR Faces Dataset

The collection application has been distributed to members of the MAPIR laboratory (http://mapir.isa.uma.es/mapirwebsite/) and their relatives to gather a new face dataset (This dataset is available at https://doi.org/10.5281/zenodo.4311764). The resulting dataset contains 1005 images from 21 individuals, from an intended number of ∼49 images per individual. These 49 images are uniformly distributed in a 7×7 grid representing the combination of the yaw space in (−65.0,65.0) and the pitch space in (−35.0,35.0) as seen in Figure 6. It is important to clarify that this face dataset is not intended to train deep neural network (DNN) due to its small size compared to existing datasets like MS-Celeb-1M or VGGFace2. Instead, it is designed as a benchmark to analyze the effect of detrimental factors due to pose variance.

The dataset is organized in the filesystem as a set of folders, each containing images from a single individual and a file in json format with information from each image:
**roll:** Roll component of the estimation in degrees.**pitch:** Pitch component of the estimation in degrees.**yaw:** Yaw component of the estimation in degrees.**rgb_image:** Name of the corresponding intensity image (from the current folder).**depth_image:** Name of the corresponding depth image (from current folder) if any is available.**bbox:** Contains the position of the face present in the image in the form of a bounding box (an array containing the [left, top, right, bottom] positions in the image in pixels). This estimated bounding box has been computed using MTCNN [43]—a common choice accompanying many state-of-the-art FR methods.


The contribution of this dataset to the wider FR community establishes a new benchmark for FR algorithms. This benchmark has been designed to gain additional insight into the performance of these algorithms in the presence of detrimental head pose conditions—one of the core problems of FR. Additionally, this kind of dataset can be used to clarify which poses are more useful to include in the gallery dataset. An example of these two use cases is shown in Section 5 and Section 6 respectively.

## 5. Evaluation on Face Recognition Algorithms

The developed dataset can be used to gain insight into the performance of FR algorithms in the presence of detrimental head pose conditions, which are not commonly represented in other evaluation datasets. However, as previously stated, a robot operating in human environments may be faced with situations showing a high variability in head poses. For this endeavor, a face identification challenge is proposed:A gallery set of known faces chosen from the complete dataset. This subset will be used as representative images for each of the individuals. Most commonly, the gallery set is composed by frontal images (center of the grid described in Section 3.2), although it can contain more images—a topic discussed more in-depth in Section 6.A query set composed by the remaining images in the dataset will be used to evaluate the FR system according to the different metrics. This process is carried out by matching the images of the query set to the most similar face images in the gallery set. This comparison commonly has the following requirements: (i) the face embeddings computed by a DNN, (ii) a comparison function, and (iii) a distance threshold used to accept or reject the match.

This process can be used to compute general measurements (e.g., recall, precision, and accuracy) and even ascertain the effects of pose variance in different FR methods (Figure 7). A comparative study between three popular open-source methods based on the state-of-the-art was carried out to both illustrate the need for such a benchmark dataset and to find out which methods are more resilient to head pose variance. Particularly, the three open-source implementations used are:FaceNet [11].
One of the most influential FR papers in recent years. It introduced fundamental concepts such as direct optimization of the embeddings and the Triplet Loss function. This optimization technique attempts to learn embeddings with smaller distances for all pairs of images of the same person (positive pairs) compared to the distance for different persons (negative pairs). This work uses a community implementation (https://github.com/davidsandberg/facenet/) of FaceNet based on Tensorflow. It provides various pretrained models, particularly a model trained on the VGGFace2 dataset [20] is used, as it is the most accurate.
ArcFace [12].
One of the more recent FR systems which shows significant performance upgrades across most common benchmarks in comparison to FaceNet. It introduces the ArcFace Loss function which, following the steps of [13,44], optimizes the angular distance between classes using a modified softmax loss function. The official implementation (https://github.com/deepinsight/insightface) uses MXNet and provides various models trained on a cleaned version of the MS-Celeb-1M dataset [21]. This work employs the provided model LResNet100E-IR.
Probabilistic Face Embeddings (PFE) [45].
A recent state-of-the-art FR approach that represents the usual face embeddings as Gaussian distributions. This method implies that some of the feature space is wasted to take into account for unreliable features such as noise, blur, and so forth—all of which can be mitigated by probabilistic embeddings. An official code implementation (https://github.com/seasonSH/Probabilistic-Face-Embeddings) based on Tensorflow is provided to accompany the paper. Particularly, this work uses the model trained on MS-Celeb-1M dataset.


These three methods have been tested using MAPIR Facesand in conjunction with their recommended comparison thresholds: ∼0.4 for FaceNet and ArcFace; and ∼−2435.04 for PFE. The gallery set is composed from the frontal head poses (pitch and yaw of ∼0.0°) contained in MAPIR Faces dataset. Using this canonical view of each user, similarity measures have been computed for all images of the same individual in the query set; i.e., there are no negative pairs. Since the algorithms may use different comparison functions and thresholds, the similarity scores have been normalized according to their thresholds to provide a fair comparison between methods. This way, if the similarity score is greater than 1.0, it is considered that the samples belong to two different individuals. Since this evaluation scheme has every possible individual in the gallery set, any image with a score greater that 1.0 is a false negative. An example of the results of these comparisons using ArcFace is shown in the Figure 7.

These similarity measurements have been compared across different yaw and pitch ranges, in Table 3 and Table 4 respectively. These results show that all 3 algorithms are affected strongly by variations in both pitch and yaw. Notably, ArcFace performs poorly when dealing with the most extreme ranges considered, as the average similarity measurements exceeds the threshold. On the other hand, FaceNet and PFE are more resilient to adverse head pose conditions.

## 6. Optimization of Face Recognition Algorithms

FR systems deployed on real environments must be provided with a gallery set composed of face images from users—whose usually depict a frontal view of the face. As shown in Figure 7, the efficacy of some methods is limited to a relatively small range of poses around the gallery image—poses further away from this radius raise type II errors (false negatives) with increased frequency. In contrast, a different gallery image can perform better on a different part of the pose space, as depicted in Figure 8. These observations suggest an increase of efficacy across the pose space the more diverse poses are included in the gallery dataset. Additionally, it raises other questions such as: whether some poses in the space are more important than others, or which amount of images to store. The inclusion of additional images also increase the computational burden on the system—since it needs to find the nearest neighbor across all images in a high dimensional space—, as well as the time needed to meet a person (i.e., to collect the required images to be stored in the gallery set), and thus a trade-off is needed.

In order to gain more insight into these questions, an optimization problem has been defined over the MAPIR Faces dataset. This problem attempts to find the most optimal sets of head poses from the whole dataset to include in the gallery set according to various FR-related metrics. Each configuration in the search space of this problem corresponds to a set of distinct poses—MAPIR Faces dataset contains 49 poses total—7 pitch variations times 7 yaw variations. The performance of a configuration is analyzed using a suite of 3 metrics, which are built upon the results reported by the FR system on the dataset:Top-1 accuracy.
It measures how many images are correctly identified by the nearest neighbors classification among the face embeddings. Additionally, the distance between the nearest pair found is thresholded according to the FR method used. This is a typical requirement in these pipelines in order to filter out unknown individuals if any exists.
Distance to nearest true pair.
It measures how well the FR system maps the face images to the deep face embeddings. Each sample embedding is compared to the same-individual embeddings in the gallery set. The resulting metric is the distance to the nearest embedding within them, this being inversely correlated to the efficacy of FR.
Distance to the nearest false pair.
It measures the separation between embeddings of different individuals in the dataset. The sample embeddings are compared to all different-individual embeddings in the gallery set. The smaller the distance between the sample embedding and the nearest different-individual embedding, the more probable it is to find false positives—particularly when the distance is inferior to the threshold.


The optimization problem in this space is an arduous task due to the dimensionality of said space. The total amount of configurations raises exponentially according to the number of poses considered. Since MAPIR Faces dataset contains n=49 distinct poses, the whole problem space has $2^{49}$ configurations. Exhaustive search for the most optimal configurations is therefore unfeasible for the confines of this work. Nevertheless, there exist some meaningful subsets of this space for which search is easier. For example, Figure 9 shows that the number of combinations of each length skyrockets in the (10,40) range, while remaining small enough to be explored at both ends of the length spectrum.

As a consequence, exhaustive search was only feasible for a small number of configuration lengths. After that point, a heuristic approaches had to be employed to search the best configurations for the defined metrics. The first heuristic approach considered is to reduce the search space by discarding combinations containing adjacent poses. This optimization relies on the assumption that the similarities of adjacent poses contains redundant information which might not be as beneficial for FR compared to more varied pairs of poses. Even after this optimization the search space remains too challenging to continue the process—being unable to effectively explore configurations past length N=10.

At this point, another heuristic approach inspired by genetic algorithms was considered to explore the remaining configuration space. This process tries to optimize one of the previously stated metrics for a certain number of poses to be considered in the gallery set. In terms of genetic algorithms [46], the heuristic search carried out has the following characteristics:An initial population is crafted from the top-30 accuracy individuals of the last population found by exhaustive search—mostly the configurations of length N=10. To fill each individual up to the required length, the remaining spots are filled by random poses.The mutation operator for each individual of length *N* can change each of the poses with a probability of 1/N. As each individual is considered a set, the poses used for the substitution are selected from the ones not already in the set.No crossover operator has been used.The most promising individuals from the population are chosen using binary tournament selection.

This way, the search space is reduced considerably compared to exhaustive search. For some combination lengths the search space contains upwards of 60 trillion configurations, whereas a genetic algorithm that generates 100 children for each of its 1000 iteration run, only explores about 100,000 combinations per combination length. Since the initial population for a configuration length (N) is based on the results of the previous search (N−1), the final results are bound to perform as well or better than the results for (N−1). The search process was carried out using ArcFace to optimize for the most accurate configurations of each length. Figure 10 depicts the evolution of the top-1 configurations according to their accuracy in the MAPIR Faces dataset. Figure 10a shows that this accuracy increase has diminishing returns in relation to their length—the top-1 configuration of length N=8 already surpasses 0.95 accuracy. Unsurprisingly, both the distance to the nearest true and false matches diminishes in relation to the configuration length (Figure 10b,c). The decrease in distance to the false match is a downside as it can raise the chance of false positives. Nevertheless, this distance stays far from the critical value of 0.4; i.e., the acceptance threshold for ArcFace.

As a result of this optimization process, we are able to list the most useful configurations in terms of accuracy as depicted in Figure 11. For instance, the best configuration of length N=3 found is shown with greater detail in Figure 12. These results can be leveraged in different ways, mainly by robots collaborating with humans in uncontrolled scenarios. As an illustrative example, let’s consider a robot operating in the headquarters of a company, where it has to recognize people’s faces in order to provide a customized HRI. In order to design a recognition system as reliable as possible, the company would register each of the known individuals in the system using the face collection application described in Section 3.2, obtaining a collection of face images from each individual uniformly distributed across the head pose space. However, in this kind of scenarios, it is common to have both time constrains to meet an employee, as well as memory/computational burden limitations. Therefore, the use of the whole pose distribution in the gallery dataset might be too burdensome, since it demands a great deal of time to meet each person, and the algorithm must find the image with the greatest similarity score among a huge amount of images. At the same time, it is also important to keep the recognition accuracy high, since recognition errors could lead to undesired situations. A robot equipped with the same FR system used in this section, i.e., MTCNN and ArcFace, can use the results reported here to find the optimal trade-off between FR accuracy and said constrains. In the case another FR system is used, the optimization process described here can be replicated with this new system, using the publicly available MAPIR Faces dataset to find the key poses to be acquired.

## 7. Discussion

This paper has presented multiple contributions towards improving resiliency of FR systems in uncontrolled scenarios. Concretely, it has explored how head pose variance detrimentally affects cutting-edge algorithms in the context of robotics applications. This endeavor was accomplished by the creation of an interactive application using state-of-the-art head pose neural networks to aid the collection of face images uniformly distributed in the space of possible head poses. HPE serves to decide which poses to save at any given time, while interactive visualizations guide the user towards the desired poses. The application is available on Github (https://github.com/samuelbaltanas/face-pose-dataset) and it can be distributed to gather images from multiple individuals. A head-pose-aware dataset has been created using said application in collaboration with the MAPIR laboratory at the University of Málaga. Coined MAPIR Faces, said dataset consists of 21 different individuals and over 1000 images. Relying on this repository, we have compared the effect of detrimental head poses on three different state-of-the-art FR networks. These results have made it clear that all 3 networks are affected by pose variance to different degrees. Particularly, ArcFace shows the most limited pose range in which it is effective, despite performing similarly or better than the other 2 in most popular benchmarks. Additionally, a method for enhancing the gallery set by collecting a more diverse poses with the aid of the previously mentioned application has been proposed. This approach takes in consideration the best poses to include for a given algorithm, reported by an optimization process over the poses in MAPIR Faces dataset. This method can be used to register users in FR applications with the advantage of storing more facial information than just including a frontal face image. It is particularly suitable for recognizing individuals in settings where the user might not look directly at the camera.

### Future Directions

The approach presented in this work leaves open multiple lines of research which should be pursued. First, MAPIR Faces dataset should be expanded, not only to gather more individuals, but also by including depth information provided by an RGB-D camera. The usage of depth information in FR is still a relatively unexplored research topic. The existence of more face datasets containing depth information may offer a way to improve existing methods. Second, the development of these FR tools for the ROS programming framework will facilitate their use in real robotic systems. Additionally, it may be beneficial to use HPE along FR at runtime to help the search for the nearest head poses—as they are also more probable to be correct matches. Finally, integration of video information over time may prove useful to enhance the accuracy of FR in robotic systems, given that they perceive the world as a continuous video feed. Current state-of-the-art FR approaches usually process video frames individually, without using information from previous frames.

There are also other avenues of research in the domain of healthcare robots and HRI worth exploring. FR enables an assistive robot to recognize previously learned individuals in its line of sight. Nevertheless, learning the identities of new individuals at runtime is a major feature for such an assistant still missing in most open-source systems—such as the ROS framework (https://www.ros.org/). Other related HRI topic that needs to be explored in more detail is the search of specific individuals, since it is a game-changing feature in many robot applications. In the context of healthcare robots, the combination of user identification and search can enable many tasks only feasible for humans. The use of mobile robots to monitor of patients condition (e.g., checking out their pulse, blood pressure, or inquiring on their physical and emotional state) is a prime example of a use case that requires robust identification and search methods. These features are also fundamental to provide a personalized user experience—from the most simple tasks, such as addressing a patient by its name; to the most sensitive ones, such as delivering a dose of medicine to the correct individual or alerting the nearest healthcare worker of an anomaly in a patient.

## Figures and Tables

**Figure 1 sensors-21-00659-f001:**
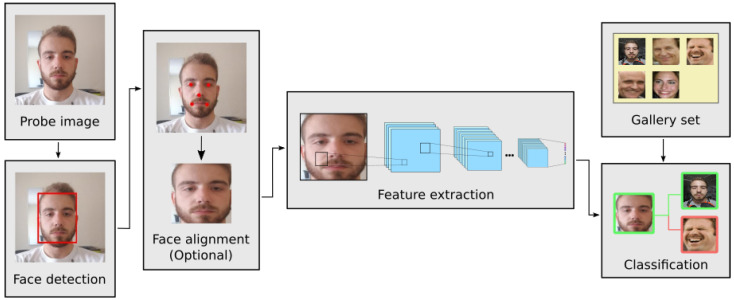
Typical pipeline for face recognition (FR) systems.

**Figure 2 sensors-21-00659-f002:**
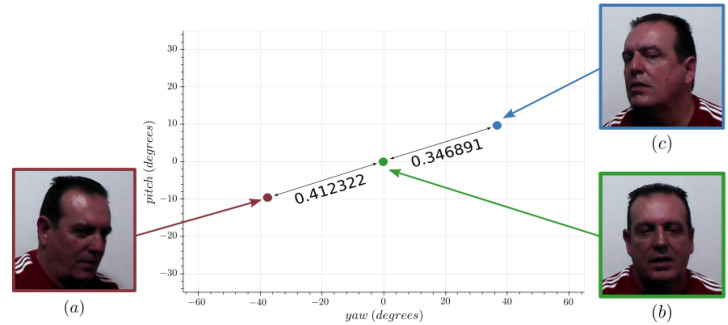
Embedding distance comparison between various poses of the same individual using ArcFace with its standard distance threshold of 0.4. The distance between (**a**,**b**) is 0.412, which is considered a negative match. On the other side, the distance between (**b**,**c**) is 0.347, which is considered a positive match.

**Figure 3 sensors-21-00659-f003:**
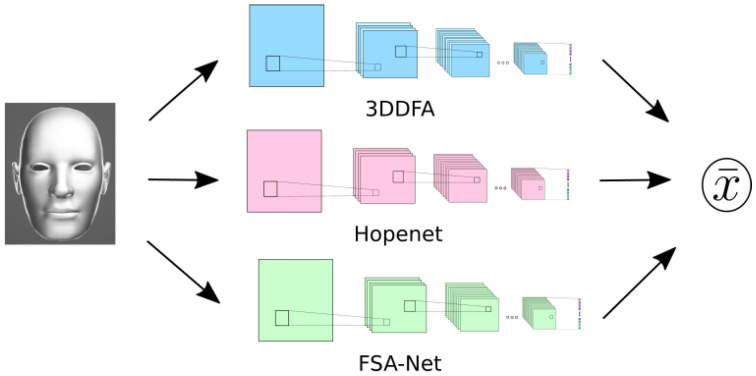
Schema of the HPE system stacking three methods, where $\bar{x}$ stands for the average of the three estimated poses.

**Figure 4 sensors-21-00659-f004:**
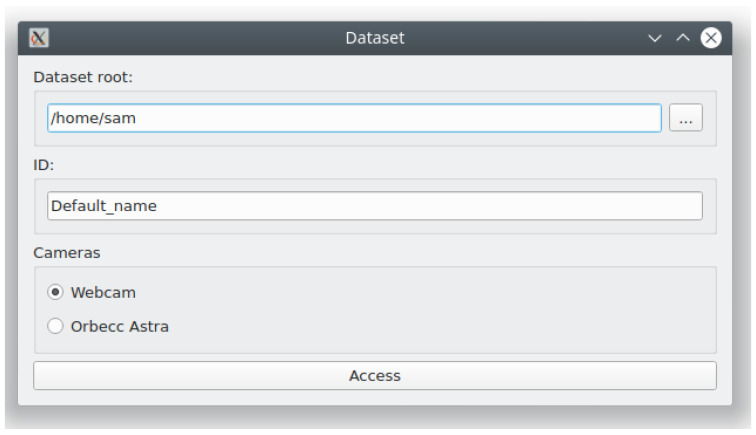
Log-in view of the developed application. It is the first window appearing when the tool is launched.

**Figure 5 sensors-21-00659-f005:**
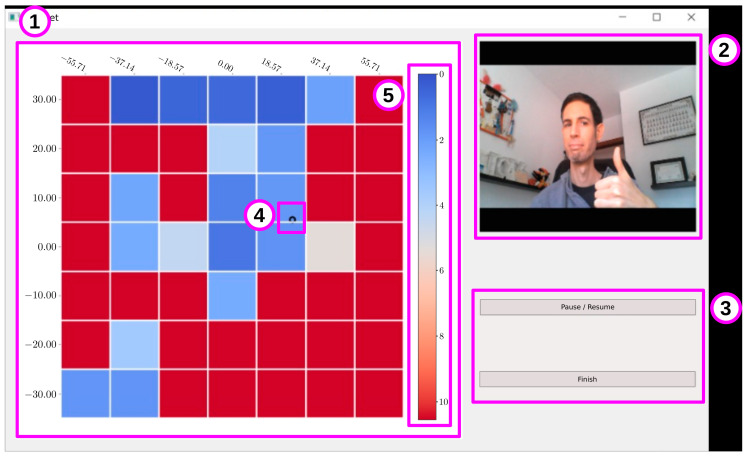
Interface of the interactive application for collecting face images. The interface is composed of 5 parts: the collection state (**①**), the camera feed (**②**), the control buttons (**③**), a black pointer (**④**), and a colorbar (**⑤**).

**Figure 6 sensors-21-00659-f006:**
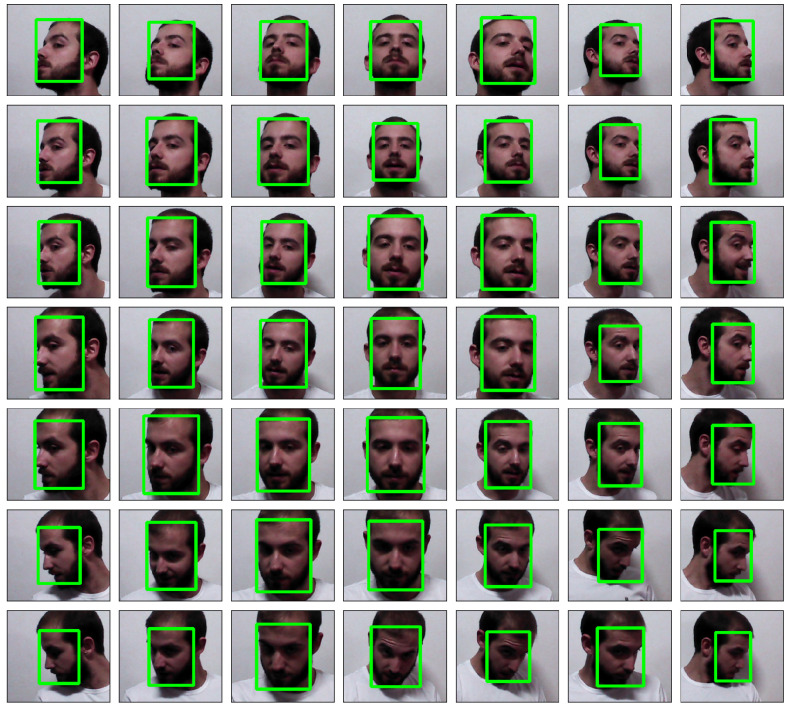
Estimated pitch and yaw results from a single individual in the MAPIR Faces dataset.

**Figure 7 sensors-21-00659-f007:**
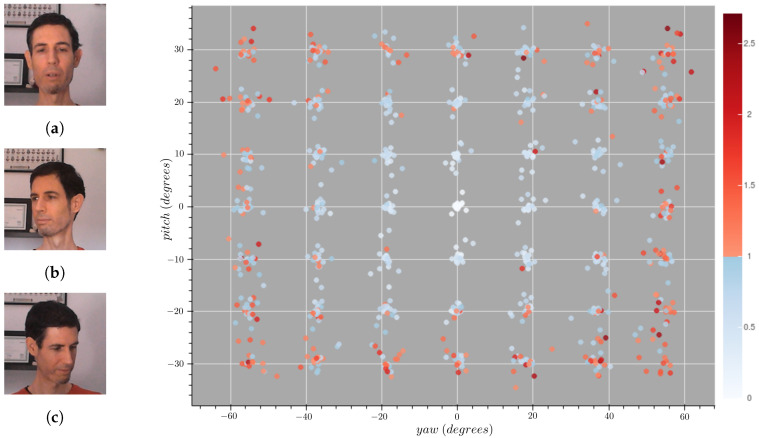
Face similarity results of ArcFace distributed across the selected head pose space. The blue samples represent correct identifications, while the red ones represent false rejections. The hue of the color is proportional to the similarity scores. The images on the left are examples of: a selected pose (**a**), a correct identification (**b**), and a false rejection (**c**).

**Figure 8 sensors-21-00659-f008:**
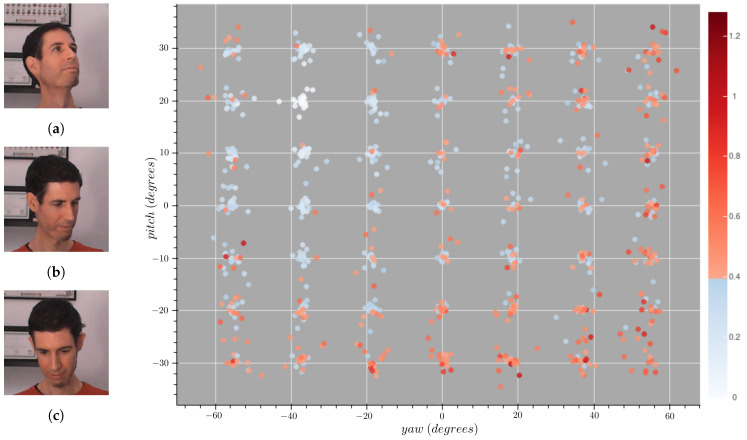
Face similarity results of ArcFace using a non-frontal face image. The gallery image depicts a head pose with an estimated pitch of 20° and a yaw of −37.2°. These samples appear in white at the top-left of the figure. The images on the left are examples of: a selected pose (**a**), a correct identification (**b**), and a false rejection (**c**).

**Figure 9 sensors-21-00659-f009:**
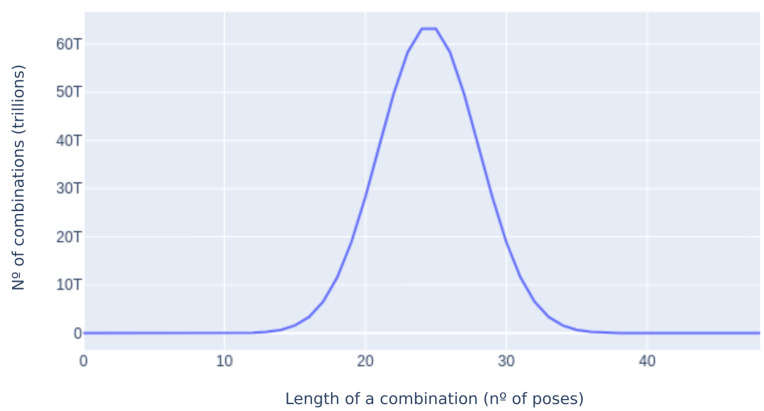
Length of a combination against the number of combinations of said length in trillions.

**Figure 10 sensors-21-00659-f010:**
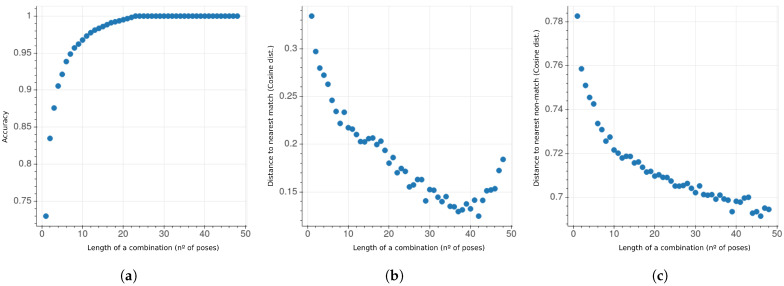
Metrics for the top-1 accuracy configurations found. (**a**) Accuracy against the number of poses. (**b**) Average distance to the nearest true match against the number of poses. (**c**) Average distance to the nearest false match against the number of poses.

**Figure 11 sensors-21-00659-f011:**
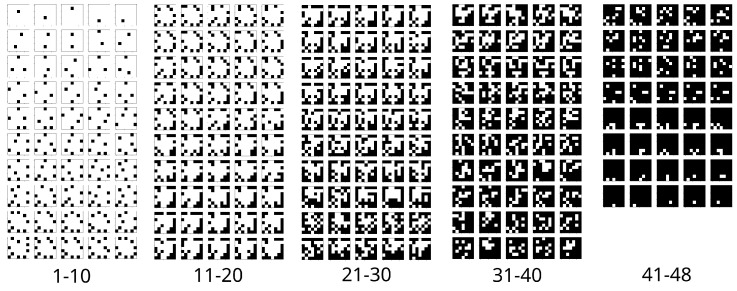
Top-5 accuracy configurations for all lengths of the gallery set. For example, the first row in the 1–10 column reports, as black squares, the 5 most optimal poses to be stored in the gallery set from a 7×7 grid of them.

**Figure 12 sensors-21-00659-f012:**
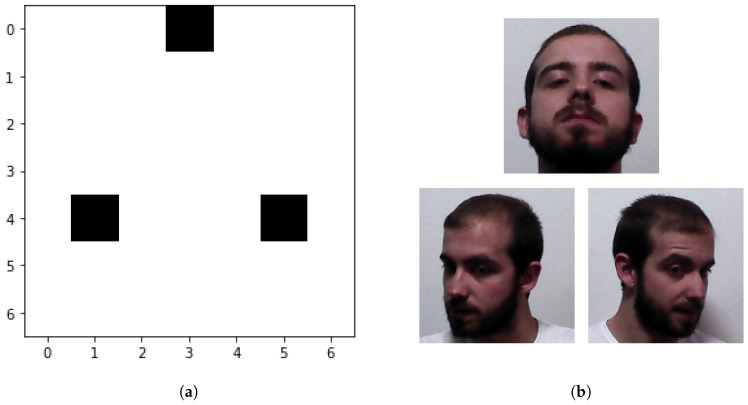
Example of the top-1 combination with 3 poses. (**a**) shows the poses in black in a 7×7 grid. (**b**) shows sample images for each of the 3 poses for an individual.

**Table 1 sensors-21-00659-t001:** Mean absolute error in degrees of the three available methods’ results on BIWI.

	Roll	Pitch	Yaw	Average
3DDFA	8.222±10.600	7.963±9.547	8.465±7.946	8.217±9.364
Hopenet	9.452±8.312	8.566±7.955	12.746±13.916	10.255±10.061
FSA-Net	7.867±7.823	11.048±8.729	15.206±17.894	11.374±11.482
Stacking	5.461±5.375	5.977±4.923	9.029±8.795	6.822±6.364

**Table 2 sensors-21-00659-t002:** Time benchmarks (mean and standard deviation in ms) for the different head pose estimators, on a Intel Core i5-9600K CPU and a RTX 2060 GPU.

	CPU	GPU
3DDFA	16.8±1.23	2.75±1.023
Hopenet	184±15.19	7.62±2.7
FSA-Net	58.3±82.87	26.4±6.83
Stacking	244±78.9	38.9±10.6

**Table 3 sensors-21-00659-t003:** Mean similarity measurements between embeddings of the same individual in relation to the yaw of the face.

Method	[0,9.3]	[9.3,28]	[28,46.4]	[46.4,65]
FaceNet [11]	0.441±0.240	0.527±0.162	0.667±0.120	0.860±0.055
ArcFace [12]	0.579±0.328	0.695±0.251	0.871±0.205	1.103±0.134
PFE [45]	0.555±0.192	0.624±0.147	0.718±0.097	0.840±0.039

**Table 4 sensors-21-00659-t004:** Mean similarity measurements between embeddings of the same individual in relation to the pitch of the face.

Method	[0,5]	[5,15]	[15,25]	[25,35]
FaceNet [11]	0.480±0.290	0.551±0.186	0.668±0.139	0.816±0.092
ArcFace [12]	0.549±0.329	0.688±0.231	0.873±0.177	1.123±0.125
PFE [45]	0.545±0.205	0.621±0.133	0.729±0.090	0.838±0.040

## Data Availability

The dataset generated during this study is publicly available at https://doi.org/10.5281/zenodo.4311764.

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
