# Peer review of "Improving the Head Pose Variation Problem in Face Recognition for Mobile Robots"

_sensors, 2021, doi:10.3390/s21020659_

Round 1

Reviewer 1 Report

Baltanas et al. present the pose variation problem while recognizing
faces. The authors backed their finding with dataset, SOTA study, and methods. The research idea is well conceived, the research design in appropriate, and the methods and sufficiently explained. However, there are some problems that need to be addressed before publication. Hence, I would like the authors to consider the comments in the next for publication. 

1- The title is not giving the complete picture of the research in a single sentence and needs to be updated.

2- Define abbreviations at the first instance of their usage in the abstract and also in the main body of the manuscript.

3- There are many grammatical and structural errors that make it difficult to understand the real meaning of some sentences. Check the manuscript with an English editor or MDPI English editing service.

4- The authors have argued in lines 15-17 that the optimized set of poses significantly improves the performance of a state-of-the-art, cutting-edge system based on MTCNN and ArcFace. However, the abstract is void of the most important findings of the present study. Also, limit the abstract within the 200 word limit for MDPI Sensors.

5- How would the authors define a reference head pose position with near 100% accuracy?

6- It would be a great idea to tag the last paragraph on page 16 as future directions for readers' understanding.

Author Response

Please see the section "Answers to Reviewer #1" in the attachment.

Reviewer 2 Report

This paper presents multiple contributions towards more resilient FR systems in uncontrolled scenarios. Concretely, it has explored how head pose variance detrimentally affects cutting-edge algorithms in the context of robotics applications.

The first remark is the following: although the abstract refers to human-robot interaction, there is no explicit reference to this interaction in the manuscript.

The second remark is the lack of proposal of a real-time facial recognition algorithm in an unconstrained environment, for example in remote surveillance.

In order to enrich the manuscript, it is imperative to cite key references in this field, in particular [1] and [2] :

[1] Masi et al., "Learning Pose-Aware Models for Pose-Invariant Face Recognition in the Wild," in IEEE Transactions on Pattern Analysis and Machine Intelligence, vol. 41, no. 2, pp. 379-393, 1 Feb. 2019, doi: 10.1109/TPAMI.2018.2792452.

[2] Adjabi, I.; Ouahabi, A.; Benzaoui, A.; Taleb-Ahmed, A. Past, Present, and Future of Face Recognition: A Review. Electronics 20209, 1188.

Author Response

Please see the section "Answers to Reviewer #2" in the attachment.

Reviewer 3 Report

This research deals with face recognition for computers. Particularly, a tool for gathering an uniform distribution of head pose images from a person, which has been used to collect a new dataset of faces and to analyse and quantifying head poses.

I would like to congratulate the authors in the works, as I find it as an exellent piece of work, well structured and of interest for the readership of the journal. Moreover, I encourage them to continue this line of research. I will be willing to know more in the near future.

My only suggestion is related to the applied level of the results, for this field or other fields. I was wondering that other fields might be interested in the current results, so my suggestion is related to increase the impact of the current manuscript. From my field, which is psychology I imagine the results can help to shed light on comparisons with human processing, such as the recognition of features, or even other fields such as Ocuppational therapy. This is just a suggestion, and authors are free to follow it if they consider it of interest.

Author Response

Please see the section "Answers to Reviewer #3" in the attachment.

Round 2

Reviewer 1 Report

The authors have significantly revised the manuscript. I believe the manuscript is now ready for publication.

Author Response

Please see the "Answers to Reviewer #1" section in the attachment.

Reviewer 2 Report

The manuscript has been significantly improved.

Author Response

Please see the "Answers to Reviewer #2" section in the attachment.
